# The Role of CDKs and CDKIs in Murine Development

**DOI:** 10.3390/ijms21155343

**Published:** 2020-07-28

**Authors:** Grace Jean Campbell, Emma Langdale Hands, Mathew Van de Pette

**Affiliations:** Epigenetic Mechanisms of Toxicology Lab, MRC Toxicology Unit, Cambridge University, Cambridge CB2 1QR, UK; gc621@mrc-tox.cam.ac.uk (G.J.C.); elh88@mrc-tox.cam.ac.uk (E.L.H.)

**Keywords:** cyclin-dependent kinase, CDK inhibitors, mouse, development, knock-out models

## Abstract

Cyclin-dependent kinases (CDKs) and their inhibitors (CDKIs) play pivotal roles in the regulation of the cell cycle. As a result of these functions, it may be extrapolated that they are essential for appropriate embryonic development. The twenty known mouse CDKs and eight CDKIs have been studied to varying degrees in the developing mouse, but only a handful of CDKs and a single CDKI have been shown to be absolutely required for murine embryonic development. What has become apparent, as more studies have shone light on these family members, is that in addition to their primary functional role in regulating the cell cycle, many of these genes are also controlling specific cell fates by directing differentiation in various tissues. Here we review the extensive mouse models that have been generated to study the functions of CDKs and CDKIs, and discuss their varying roles in murine embryonic development, with a particular focus on the brain, pancreas and fertility.

## 1. Introduction

Cyclin-dependent kinases (CDKs) are proteins that, by definition, require the binding of partner cyclin proteins in order to phosphorylate a series of target proteins. These complexes are required for appropriate progression of the cell cycle and, as such, are essential for development. CDK inhibitors (CDKIs) are a class of proteins that preferentially bind to and inhibit the activity of CDK/cyclin complexes and are thus involved in stopping or delaying the progression of the cell cycle. The roles and functional redundancies of CDKs and CDKIs in regulating the cell cycle have been extensively reviewed previously [1,2,3]. The structures of these proteins, including potential sites for interactions with inhibitors, have also been extensively studied and reviewed [4,5,6,7,8].

Twenty CDKs have been described in the mammalian genome, derived from a single ancestral gene in yeast, and many of these have been shown to be expressed during embryonic development [9,10]. Likewise, there are two major families of CDKIs: the CIP/KIP family, encoded by the *Cdkn1* gene family comprising p21^CIP1^, p27^KIP1^ and p57^KIP2^; and the INK4 family, encoded by the *Cdkn2* gene family comprising p16^INK4a^, p15^INK4b^, p18^INK4c^ and p19^INK4d^. Many CDKIs have been shown to play key roles in embryonic development [11,12,13]. In addition to their shared ability to modulate the cell cycle, several of these CDKs and CDKIs have been shown to direct differentiation of a number of different cell types with a particular convergence in the brain, pancreas and gonadal ridges, which will be discussed in greater detail later on. 

For many of these proteins, their functions have been studied primarily through knock-out (KO) mouse models and cell lines. These are mice or cells with one or both alleles of the gene in question constitutively deleted from the genome with any phenotypic abnormalities ostensibly due to the absence of the specified protein. In the case of CDKs and CDKIs, family members often display redundant functions, and as a result, many of the expected phenotypes that might be caused by the absence of a family member can be compensated for by other members, or even evolutionarily distant homologues, despite low sequence conservation [14,15]. To elucidate the extent of this compensation within gene families, researchers have frequently generated double (dKO) and triple (tKO) KO mouse lines of selected family members. Additionally, conditional KO (cKO) models can be used where the gene deletion is triggered through *Cre-recombinase* activity, allowing the deletion to be tissue-specific or induced at a desired time point. In this way, cKO animals can provide additional information about a protein’s function by focussing on a particular developmental area and are particularly useful for bypassing a constitutive KO’s embryonic lethality.

This review aims to summarise the known functions of CDKs and CDKIs in murine development, with a particular focus on development of the brain, pancreas, sperm and oocytes, principally using information gained from mouse models. The phenotypes of mouse CDK and CDKI KOs have been previously reviewed and an updated profile is provided here with a focus on development (Table 1 and Table 2) [3,16,17,18,19,20]. The primary organs affected by each CDK and CDKI, as analysed through mouse KOs, are shown schematically in Figure 1, with KOs of *Cdk1*, *Cdk4*, *Cdk6*, *Cdk11*, *Cdk13*, *Cdkn1b*, *Cdkn1c* and *Cdkn2c* affecting overall embryonic development.

## 2. Absolute Requirement of *Cdk1*

*Cdk1* is essential for successful completion of the cell cycle [18,21], preventing the creation of constitutive *Cdk1^−/−^* cell lines or mouse models. Attempts to create *Cdk1*^−/−^ offspring result in immediate failure following fertilisation, due to the inability to successfully complete cell division. This requirement cannot be compensated for by other members of the CDK family [84] and is even observed in a mutant line with cassette insertion resulting in heterozygous cells expressing only 50% of wild-type levels of *Cdk1,* which also results in early embryonic lethality [22]. It is therefore perhaps surprising that such a requirement for successful completion of the cell cycle is not observed in any of the other CDK family members.

*Cdk1* is also indispensable for cell cycle completion during gamete development [85]. It is essential for the formation of sperm, as *Cdk1* cKO spermatocytes result in cell arrest at prometaphase and thus male infertility [23]. Additionally, *Cdk1* cKO oocytes lead to female infertility as the oocytes are unable to resume meiosis, with a microinjection of Cdk1 rescuing the phenotype, thus confirming the necessity of *Cdk1* in mouse female fertility [24]. Cdk1 maintains phosphorylation of protein phosphatase 1 and lamin A/C for meiosis resumption in the oocytes [24], and is necessary as one of the factors required to phosphorylate eIF4E-BP1, a protein involved in facilitating spindle formation and stability during meiosis in oocytes [86]. Cdk1 phosphorylation is believed to be particularly required for resumption of meiosis in oocytes, with a natural increase in Cdk1 levels in later life resulting in speeding up meiosis I, thus potentially leading to errors such as aneuploidy in later-life pregnancies [87]. This can be ameliorated by inducing excess cyclin B1, as exploitation of preferential destruction of unbound B1 results in sustained Cdk1 levels, prolonging the cell cycle, and thus reducing aneuploidy errors [88]. Other members of the CDK family, in particular *Cdk2*, are also required for fertility, as discussed below.

## 3. Embryo

We would predict that loss of expression of any of the CDK family members would result in impaired embryonic growth, due to disruption in the normal pathways of the cell cycle. Indeed, this trend is largely followed, with *Cdk4* and *Cdk**6* null animals displaying reduced embryonic growth [27,29,31,41,89], and *Cdk11*^−/−^ embryos failing to progress beyond day E4.5 [44]. While *Cdk4* is not absolutely required for embryonic viability, *Cdk4^−/−^* animals are born at a reduced frequency from heterozygous matings, suggesting partial embryonic lethality [29]. *Cdk3*, however, appears to be functionally redundant, as a null point mutation was not found to affect embryonic development [28].

While single KOs of *Cdk2*, *Cdk4* and *Cdk6* are viable, this may be due to functional redundancy as tKO mice without *Cdk2*, *Cdk4* and *Cdk6* begin dying at day E13.5 of gestation, and have malformed livers and hearts exhibiting thinner ventricular walls than wild-type mice [21]. It is thought that loss of *Cdk4* in particular is key for the embryonic lethality of the tKO, with *Cdk2* and *Cdk6* dKO mice being viable but exhibiting a combination of the phenotypes seen in *Cdk2* and *Cdk6* single KOs, such as reduced body size and haematopoietic defects [41].

A functional relationship between *Cdk2* and *Cdk4* remains unclear, with one study showing that a *Cdk2* and *Cdk4* dKO also results in embryonic death at around day E15, due to heart defects [90]. In contrast, another study found that *Cdk2* and *Cdk4* dKO mice complete embryonic development and are born, slightly underweight, but die shortly after birth, potentially also due to heart failure [91]. *Cdk4* KO mice with a *Cdk2* ERT cKO however, show similar developmental phenotypes to *Cdk4* KO mice [91].

*Cdk4* and *Cdk6* dKO mice have altered haematopoiesis resulting in severe anaemia, with embryos being undersized and beginning to die from day E14.5, with pups that survive to birth dying soon afterwards [41]. A partial dKO where one allele of *Cdk4* is expressed but *Cdk6* is knocked-out results in viable offspring, while expression of one allele of *Cdk6* and complete *Cdk4* KO results in partial embryonic lethality, again suggesting that *Cdk4* has the more important role in embryonic viability [41].

In contrast to *Cdk2, Cdk4* and *Cdk**6*, which all display some degree of functional redundancy, deletions of *Cdk5, Cdk11* and *Cdk**13* display variable levels of embryonic lethality in isolation [32,44,45]. *Cdk5* expression is largely restricted to the brain, with constitutive and neuron-specific deletion of the gene resulting in overlapping phenotypes and perinatal lethality (~60%) [32]. *Cdk11^−/−^* blastocysts are terminated around day E3.5 due to a massive failure in mitosis [44], while *Cdk13^−/−^* embryos fail by day E16.5 [45]. For the remaining CDKs, a number remain untargeted, restricting our knowledge of their potential impact on embryonic growth parameters.

The data generated from CDK mouse models largely corresponds with the presumed function of promoting embryonic growth. Conversely, we would expect that expression of CDKIs will consequently restrict embryonic growth, due to negative regulation of the cell cycle. It is therefore surprising that although control of the cell cycle at G1/S phase was impaired in the absence of *Cdkn1a* (p21^CIP1^) [47], most likely through a general reduction in inhibition of CDK activity, the mouse model did not present with altered embryonic growth parameters. This finding is in stark contrast to the other members of the CIP/KIP family of CDKIs. Indeed, *Cdkn1b^−/−^* mice, which lack p27^KIP1^, are oversized due to increased cell proliferation, and exhibit proportionally larger internal organs [52,53,54]. This phenotype was not accompanied by reduced embryonic survival. However, the loss of control of the cell cycle did present with postnatal tumour formation [54]. It has been noted that many of the functions of *Cdkn1b* and *Cdkn1c* are functionally redundant, with a dKO line displaying greatly enhanced phenotypes of the single KOs [92]. tKO mice lacking all *Cdkn1* family members develop normally until mid-gestation, dying around day E13.5, marginally earlier than *Cdkn1b*/*Cdkn1c* dKO mice at day E15.5, with both *Cdkn1a*/*Cdkn1b* and *Cdkn1a*/*Cdkn1c* dKO mice surviving past this point [93]. Knock-in of p27^KIP1^ to replace p57^KIP2^ results in viable and healthy mice, with only a few developmental defects, suggesting p27^KIP1^ and p57^KIP2^ have overlapping but not identical developmental roles [94]. Both p27^KIP1^ and p57^KIP2^ appear to be expressed frequently in postmitotic cells, with the role of p57^KIP2^ restricted to organogenesis, as expression is not maintained into adulthood, while p27^KIP1^ is expressed both during development and in adult mice [95].

Individually, *Cdkn1c* must be viewed in a slightly different light to its family members, as this gene undergoes genomic imprinting. The paternally inherited allele of this gene is epigenetically silenced throughout the mouse’s life-course, with functionally monoallelic expression from the maternally inherited allele [96]. To the best of our knowledge, none of the other CDKs or CDKIs have been shown to undergo this process. As a result of this, a heterozygous deletion of just the paternal allele (silenced) results in wild type expression levels of the gene, while a deletion of the maternal allele (active) presents as functionally homozygous-null. Independent deletions of this gene have shown near total late perinatal or very early neonatal lethality [64,65,67,97]. The reasons for this lethality remain unclear, as although mice present with a cleft palate, preventing feeding and causing breathing difficulties [63,64,65], this would not explain the large degree of perinatal lethality that is observed. Likely the cleft palate represents one of a number of defects that combine to cause lethality, alongside abdominal wall and heart defects. Due to the role of *Cdkn1c* in blocking the cell cycle, it would be predicted that null animals would be overgrown. However, no such phenotype is observed at birth [63,64,65,68,98]. Instead, researchers have found a complex picture, where early embryonic overgrowth is lost in the late stages of a pregnancy, due to a restriction in growth potential in part attributable to the size of mouse litters [68]. Partly mirroring these results, increased dosage of *Cdkn1c* through the use of bacterial artificial chromosomes (BAC) has found an exquisitely dosage-sensitive embryonic growth restriction phenotype [99].

While a plethora of work has examined the roles of the CIP/KIP family in embryonic growth and development, the function of the INK4 family in the embryo remains comparatively untested, with studies predominantly focussed on postnatal consequences and tumour formation following deletion of a family member. Loss of *Cdkn2c* (p18^INK4c^) results in giantism and organomegaly, with the organomegaly phenotype increased upon dKO with *Cdkn1b* (p27^KIP1^) [79]. Both *Cdkn2b* (p15^INK4a^)/*Cdkn2c* (p18^INK4c^) and *Cdkn2c* (p18^INK4c^)/*Cdkn2d* (p19^INK4d^) dKO mice are viable with no morphological or behavioural abnormalities at birth [78,100]. While we cannot assume that this is the extent of the roles for the INK4 family in the regulation of embryonic growth, it must be remembered that this family of proteins is specific in its activity towards Cdk4/6 complexes, while the CIP/KIP family displays a more general inhibition of CDK family activity [101], potentially differentiating the families with regards to embryonic requirements.

## 4. Brain Development and Function

Functional roles for CDKs in brain development have so far been restricted to just a handful. While it is essential for embryonic development and assumedly in brain development, *Cdk1* does not appear to have a postmitotic role in neurons, with a cKO in such cells resulting in normal development [25]. Further testing of a *Cdk1* role in different regions of the brain remains unfeasible outside of postmitotic cells.

*Cdk5* has been shown to be essential in mouse brain development, with cKOs indicating a variety of functions in various regions of the brain, overcoming the embryonic lethality observed in constitutive KOs. A constitutive KO of *Cdk5* results in altered brain development, with more than 60% of developing embryos dying in utero [32]. cKOs of *Cdk5* do not result in the same embryonic lethality, but still cause behavioural issues [34,35,36]. The dosage and activity of *Cdk5* is tightly regulated in the brain [102]; in particular, p35 and p39 are both essential for Cdk5 activity and may be the principal activators of *Cdk5* [103]. *Cdk5* is involved in dendritic spine formation, is important in Purkinje cell migration and dendritic growth, and is required for formation of ventral striation; loss of *Cdk5* causes GABAergic neuronal death in the developing mouse foetus, as shown in *p35* cKO and *p39* KO mice [104,105,106]. Cdk5 phosphorylation of the neural protein Synapsin III is needed to regulate neurite outgrowth and axonal elongation [107]. Cdk5 phosphorylation is also required for the membrane targeting of, and promotion of proteasome-dependent degradation of Cx43, a protein involved in neuron migration and embryonic brain development, which is associated with adult anxiety-related behaviour [108].

Loss of *Cdk5* specifically in CNP cells affects myelination and results in impaired learning and memory [36,37]. CaMKII *Cdk5* cKO, which results in deletion in the mouse forebrain, causes many behavioural abnormalities, seizures, and, intriguingly, increased susceptibility to cocaine [34,35], while absence of *Cdk5* in the mid and hind-brain or in *Emx1*-expressing cells, which are involved in corticogenesis, results only in morphological defects with no reported behavioural changes [38,39,40]. It is interesting to note that despite none of the CDKIs having a specific affinity for *Cdk5*, it is the only CDK to have overlapping expression patterns in the brain with CDKI family members.

All three members of the CIP/KIP family have described roles in neural development and regulation. Interestingly, while the role of *Cdkn1a* appears to exclusively revolve around its ability to regulate cell cycle progression [48], *Cdkn1b* and *Cdkn1c* both perform this role, and in addition to this, are responsible for directing differentiation of a number of neural cell types [58,59,60]. *Cdkn1b* and *Cdkn1c* are expressed transiently in the brain during development [95], with *Cdkn1c* being expressed in the brain at day E11.5-E13.5 [109]. *Cdkn1c* has been shown to suppress growth with a constitutive KO resulting in macrocephaly due to an increased number of cortical precursors entering the cell cycle, which combines with a shortened cell cycle length and leads to an increase in cortical surface, thickness and cell numbers [60]. However, this impact has recently been shown to only occur non-cell-autonomously, with neuronal cell-specific KOs identifying a cell-autonomous role of *Cdkn1c* that promotes growth and cell survival [110]. Mosaic analysis with double markers technology was used to show that this cell-autonomous role is dependent on the existence of the locus, regardless of parental origin or transcription [110]. This role is further solidified by the observation that overexpression of *Cdkn1c* in embryos is associated with a reduction in the number of cortical progenitor cells leading to a decrease in upper layer neurons in the developing brain [111]. Finally, a recent study has hinted at a possible role for the paternal allele of *Cdkn1c* in neocortical development, despite this gene being paternally imprinted and therefore not transcribed [69].

An oligodendrocyte disorder resulting in loss of white brain matter links increased progenitor cell proliferation and increased differentiation with increased *Cdk2* and decreased p21^CIP1^ and p27^KIP1^ activity [112]. Additionally, impairment of potassium channels in the brain related to seizure disorders was reversed by decreased *Cdk2* or increased p27^KIP1^ activity [113]. In wild-type embryos, p27^KIP1^ expression is found to occur along a gradient in developing neural tubes, with the highest expression in postmitotic cells of the neuroepithelium, with the deletion of a component of the nuclear pore complex showing a correlation between altered p27^KIP1^ expression patterns and abnormalities in cell proliferation during development [114].

The INK4 CDKI family does not appear to have an important role in the developing mouse brain, with no known studies showing a requirement for any of the *Cdkn2* genes. Indeed, even a dKO of both *Cdkn2c* and *Cdkn2d* appears to result in no neurological defects [100].

## 5. Pancreas

Both CDKs and CDKIs appear to have a role in insulin regulation, as changes in their expression affect pancreatic development, with a particular functional convergence in islets. *Cdk4* in particular is vital to pancreatic function as a *Cdk4^−/−^* model results in mice with greatly impaired insulin secretion [29,30]. In contrast, a *Cdk8* cKO specific to β-cells increases insulin secretion and thus improves glucose tolerance [43]. However, these cells have lost their ability to deal with stress, leading to increased apoptosis under oxidative stress [43]. Phosphorylation of Ngn3 by CDKs, including Cdk1 and Cdk2, controls differentiation of endocrine cells in the embryonic pancreas and is required to maintain adult β-cell function [115]. The importance of CDKs in controlling the pancreatic tissue cell cycle is attracting pharmaceutical interest, with CDK inhibitors being shown as potentially viable novel approaches to treat pancreatic adenocarcinomas [116].

The specific roles of CDKIs in pancreatic development are less clear. A dKO of *Cdkn2c* and *Cdkn2d* causes hyperplasia of pancreatic β-cells [100], but no other role for the INK4 family in the pancreas has yet been identified. KOs of members of the CIP/KIP family do not appear to greatly affect the developing pancreas, but their expression is still evident. For instance, upregulation of *Cdkn1a* in the pancreas is associated with a decrease in islet cell proliferation [117]. While p57^KIP2^ expression has been observed in the pancreas, most notably at day E13.5 and then rapidly declining [109], p27^KIP1^ does not appear to be expressed [95]. Rat p57^KIP2^ is highly homologous to mouse p57^KIP2^, while both differ from human p57^KIP2^ [118]. Similar to mice, rat p57^KIP2^ expression is tightly controlled in the pancreas, decreasing at birth and increasing again two weeks after birth, but is not evident in β-cells [118]. A complex picture has emerged for *Cdkn1c* whereby the gene indirectly regulates islet mass and β-cell populations [119]. Overexpression models have found newborn mice to have significantly altered blood glucose profiles, which are then retained through to adulthood [120]. Some of the exact nature of this role remains unclear, in part due to the difficulty of dissecting the functions of *Cdkn1c* from those of surrounding genes in the *Kcnq1* imprinting cluster, with further studies needed to elucidate these specific roles.

## 6. Fertility

Many CDK KO mouse models result in reduced fertility or even complete infertility, highlighting essential roles for CDKs in mouse fertility. *Cdk2*, *Cdk4*, *Cdk16* and *Cdk18* have all been found to be expressed in developing ovaries [121]. *Cdk2* is isolated to premeiotic spermatocytes in the testes, while *Cdk16* and *Cdk18* are expressed in postmeiotic spermatids, suggesting diverse roles in male fertility [121]. In particular, *Cdk2* is dispensable for growth but not fertility [122]. A *Cdk2* KO causes both male and female infertility, due to both spermatocytes and oocytes arresting at prophase I [26,27,85]. However, an oocyte-specific *Cdk2* cKO resulted in normal oocyte development, with the authors concluding that *Cdk1* was the only family member required for resumption of meiosis [24]. *Cdk2* is also required for differentiation of spermatocytes, with a loss-of-function allele resulting in increased apoptosis of gonocyte precursors following depressed differentiation into spermatogonial stem cells [123]. A *Cdk2* and *Cdk6* dKO also results in infertility in both sexes, while a *Cdk4* and *Cdk6* dKO displays the same infertility seen in a *Cdk4* single KO, with total female infertility and partial male infertility [29,30,31], signifying that *Cdk6* does not have a dominant phenotype in mouse fertility [41]. The impaired spermatogenesis in *Cdk4* KOs appears to be due to defective proliferation of gonocytes and hypoplastic seminiferous tubules resulting in testicular atrophy [30]. A lack of *Cdk4* causes female sterility through suppressing development of the pituitary, resulting in reduced lactotrophs and a deficit in the hypothalamic–pituitary axis, leading to perturbed corpus luteum formation [29,30,31]. Cdk16 interacts with, and is phosphorylated by, cyclin Y-like1 to regulate spermatogenesis [89]. A *Cdk16* cKO resulted in structurally malformed and consequently hypomotile spermatozoa, leading to reduced male fertility, placing *Cdk16* function in the terminal stages of spermatogenesis [46]. *Cdk16* does not appear to be required for female fertility [46]. A more in-depth review of the role of CDKs in spermatogenesis has been compiled by Palmer et al. [124].

A lack of *Cdkn1b* (p27^KIP1^) results in female sterility due to impairment of ovarian follicle development and corpora lutea formation, believed to be related to the oversized pituitary and thus alteration of the hypothalamic–pituitary–ovarian axis which is seen in these KO mice [52,53,54]. Despite having overly large testes, male fertility was not affected in *Cdkn1b* KO mice [54]. Loss of *Cdkn1c* (p57^KIP2^) has been shown to result in sexual immaturity in both sexes [65]. Decreased *Cdkn1a* and *Cdkn1c* expression in the testes is associated with mitotic arrest in the male germ cells [125]. *Cdkn1c* expression is observable in rat testes at day E17, declines at day E19.5 to day E21.5, but is again evident in spermatogonia from the second week after birth and in adult spermatids [118]. In mice, while *Cdkn1b* expression has been observed in the ovaries and testes, *Cdkn1c* does not appear to be expressed [95].

Contrastingly, loss of *Cdkn2a* (p16^INK4a^), *Cdkn2b* (p16^INK4b^) or *Cdkn2c* (p18^INK4c^) does not appear to affect fertility [70,78]. Additionally, mice without both *Cdkn1b* and *Cdkn1c* remain fertile [78]. Loss of *Cdkn2d* (p19^INK4d^), however, decreases but does not abolish male fertility, due to testicular atrophy and increased apoptosis of male germ cells [81]. Similarly, a dKO of both *Cdkn2c* and *Cdkn2d* produces fertile females but sterile males [100]. These dKO males exhibit small testes with delayed progression through meiosis leading to increased germ cell apoptosis and decreased sperm cell count with malformed existing spermatozoa, suggesting both p18^INK4c^ and p19^INK4d^ are required to regulate differentiation of mature sperm [100]. Alternative splicing of *Cdkn2c* leads to two distinct mRNAs, the shorter of which is found in various differentiated cell types such as muscle, lung and liver, while the longer mRNA is isolated to the testes in undifferentiated, mitotically active cells, suggesting that the two protein isotypes have separate roles, particularly for differentiation [126]. p15^INK4a^, p16^INK4b^, p21^CIP1^ and p27^KIP1^ are all expressed in the ovary, but p16^INK4b^ is expressed more highly than the other CDKIs and seems to decrease during oocyte growth [127].

## 7. In Vitro Studies

*Cdk8* and *Cdk19*, as involved in the CDK–Mediator complex, are vital for regulating gene expression following differentiation of embryonic stem cells into neuronal precursors [128]. *Cdk8* in particular is needed to bind to distal enhancers to promote gene activation upon differentiation [128]. In the absence of Jdp2, p21^CIP1^ is required to bind to antioxidant response elements of the Slc7a11 promoter in order to provide redox control and block random oxygen species mediated apoptosis in granule cell progenitors, an essential component for normal development of the cerebellum [129].

*Cdk16* is required for dendrite development and is regulated by Cdk5 phosphorylation [130]. Cdk16 binds to p35 and is phosphorylated by Cdk5 to enhance its own kinase activity [33]. Additionally, phosphorylation of Cdk16 by BRSK2 reduces glucose-stimulated insulin secretion in mouse β-cells [131].

## 8. Discussion and Future Perspectives

While a great deal of data has been generated on the roles of both CDKs and CDKIs in murine embryonic development, as has been presented in this review, there remains a number of as yet entirely untested gene family members. Much of the focus of research into these families will remain in cancer biology, given the oncogenic (CDKs) and tumour suppressing (CDKIs) nature of the groups, respectively. There is a wealth of data being generated on the use of CDK inhibitor compounds (CDKis) and their efficacy for tumour treatment [4,132]. While developmental biology and tumour therapy seem like divergent focuses, much of the data that has been generated in embryonic development can serve as a therapeutic guide. Thanks to multiple knockout lines and environmental studies, researchers have uncovered shared and redundant functions of both CDKs and CDKIs. This data will be invaluable in future determinations of appropriate CDKi therapies, based on the expression profiles of the CDK and CDKI families in tumours.

Further study of *Cdk1* will remain challenging due to its critical role in cell cycle progression. However, researchers have demonstrated that this absolute requirement can be bypassed in certain scenarios using conditional deletions, which have also been utilized to dissect the multiple roles of *Cdk5* in brain development. There remains the opportunity to target *Cdk11* and *Cdk13* in a similar manner, so as to bypass the total embryonic lethality seen in the respective constitutive knockout mouse lines and enable cell type specific functional role analyses. Beyond these, a number of the CDKs are awaiting their functional roles to be examined, as has been highlighted in Table 1.

The CDKIs present with a similar picture, where some family members have been studied extensively through both constitutive and conditional deletions, while others remain to be tested. Perhaps surprisingly, only *Cdkn1c* has been used to explore the impact of overexpression of a CDKI [99,120]. These lines have shown an exquisite dosage sensitivity of the gene on both embryonic growth and tissue differentiation. Given the known shared functional roles of the CIP/KIP family [92,93,94,95], *Cdkn1a* and *Cdkn1b* appear to be ideal candidates for a similar targeting strategy and further exploration of the shared and unique roles of these family members. Recently, researchers have begun applying a novel approach to the study of these genes, through the generation of bioluminescent reporter lines. *Cdkn2a*, *Cdkn1b* and *Cdkn1c* have all been independently targeted with a luciferase-reporter to monitor endogenous gene expression in vivo [133,134,135]. While these lines were initially largely developed to enable in vivo tumour formation, progressive technological development has allowed sensitive high-resolution reporting of embryonic *Cdkn1c* expression in utero. This approach uncovered a previously unseen sensitivity of the epigenetic markers that regulate this gene’s expression to gestational dietary modification that resulted in lifelong misexpression of the gene. As a read-out for gene expression, these models offer unparalleled insights, due to the ability to image individual animals throughout their life-course, and therefore generate a complete picture of expression. It is easy to envisage this approach being applied to any of the CDKs or CDKIs to explore as yet unknown functions and provide a complementary addition to the catalogue of presently assembled mouse models, which will continue to expand and develop our knowledge of these gene families.

## Figures and Tables

**Figure 1 ijms-21-05343-f001:**
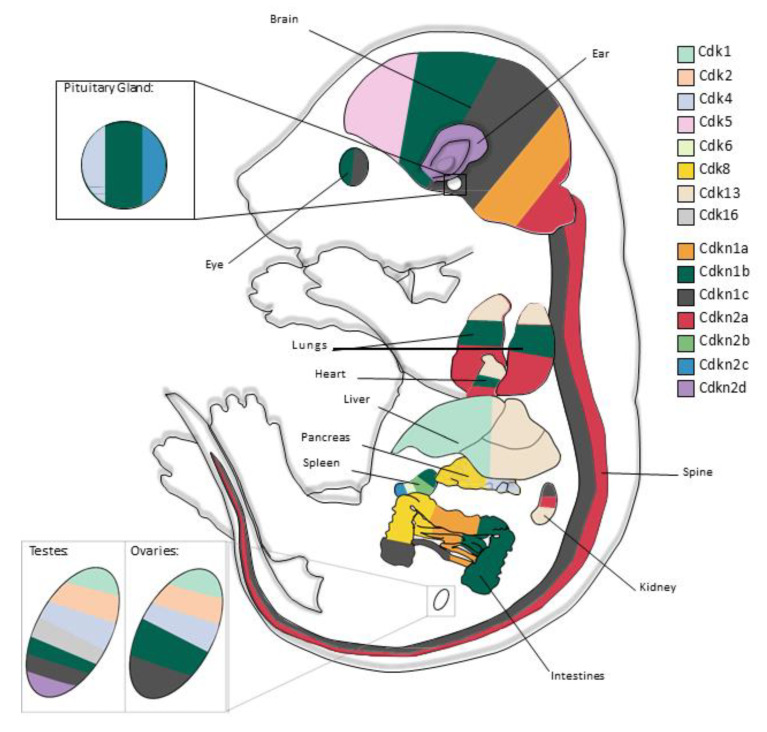
Schematic of principal areas of CDK and CDKI involvement during murine development, as determined by KO and cKO mouse models. The ears, brain, pituitary gland, eyes, spine, lungs, heart, liver, pancreas, kidneys, spleen, intestine and testes/ovaries are colour-coded based on the presence of an altered phenotype upon KO of a CDK or CDKI, depicted here in an E17.5 mouse embryo. The area of each colour is not reflective of the magnitude of the effect on the mouse phenotype. KOs of *Cdk1*, *Cdk4*, *Cdk6*, *Cdk11*, *Cdk13*, *Cdkn1b*, *Cdkn1c* and *Cdkn2c* affect overall embryonic development.

**Table 1 ijms-21-05343-t001:** Phenotypes of CDK (cyclin-dependent kinase) KO (knock-out) and cKO (conditional KO) in mice.

CDK	Developmental Region	KO Phenotypes	References
*Cdk1*	Whole Body	Mutant line with truncated protein–homozygous early embryonic lethal	[21]
	Whole Body	KO–lethal at the 1-cell stage due to inability to complete cell division	[22]
	Testes	cKO in spermatocytes–cells arrest at prometaphase resulting in male infertility	[23]
	Ovaries	cKO in oocytes–females infertile due to failure to resume meiosis. Rescued with microinjection of Cdk1	[24]
	Brain	cKO in Nex (postmitotic neurons of cortex and hippocampus)–normal brain development	[25]
	Liver	cKO in liver–failure to complete cell division but normal phenotype due to compensatory cell growth	[22]
*Cdk2*	Whole Body/ Testes/Ovaries	KO–develop normally but both sexes sterile. Spermatocytes and oocytes arrest at prophase I, oocytes slightly later than spermatocytes	[26]
	Whole Body/ Testes/Ovaries	KO–normal development but both sexes infertile	[27]
	Ovaries	cKO in oocytes–develop normally	[24]
*Cdk3*	Whole Body	Null point mutation–normal development. Possible functional redundancy	[28]
*Cdk4*	Whole Body/ Testes/Ovaries/ Endocrine System/ Pancreas	KO–reduced mutant numbers in heterozygous cross, suggesting some embryonic lethality. Decreased body size. Defective spermatogenesis reducing fertility. Females infertile due to deficit in hypothalamic-pituitary axis. Mice are insulin-deficient diabetic due to affected β-cells.	[29]
	Whole Body/ Pancreas/ Testes/Ovaries	KO–mice are undersized and develop diabetes as above. Males show testicular atrophy due to defective proliferation of gonocytes and hypoplastic seminiferous tubules. Female infertility due to perturbed corpus luteum formation	[30]
	Whole Body/ Endocrine System	KO–decreased body size with small pituitaries, with particularly reduced lactotrophs resulting in female infertility	[31]
*Cdk5*	Brain	KO–>60% die in utero; altered brain development	[32]
	Brain	KO–reduced phosphorylation of histone H1 by Cdk16 in the brain	[33]
	Brain	cKO in CaMKII (forebrain)–seizures, tremors, growth retardation, layer disruption and neurogenerative changes that progressed with age	[34]
	Brain	cKO in CaMKII–behavioural changes, increased susceptibility to cocaine	[35]
	Brain	cKO in CNP cells–hypomyelination and impaired learning and memory	[36]
	Brain	cKO in CNP cells–loss of myelin repair	[37]
	Brain	cKO in EMX1–impaired differentiation of oligodendrocytes	[38]
	Brain	cKO in EMX1–defective layer structure in cerebral cortex	[39]
	Brain	cKO in mid and hindbrain– small cerebellum and altered dendritic development	[40]
*Cdk6*	Whole Body/ Immune System	KO–undersized females. Underdeveloped thymus and spleen. Mild anaemia due to defects in haematopoiesis	[41]
*Cdk7*			
*Cdk8*	Whole Body/ Intestine	KO–normal growth, but increased size and growth rate of intestinal tumours	[42]
	Pancreas	cKO in β-cells–improved glucose tolerance due to increased insulin secretion. Increased cell death following stress conditions	[43]
*Cdk9*	Whole Body	KO–complete embryonic lethality from day E9.5	*
*Cdk10*			
*Cdk11*	Whole Body	KO–early embryonic lethal between days E3.5 and E4. Cells exhibit growth defects and mitotic arrest, resulting in blastocyte apoptosis	[44]
*Cdk12*			
*Cdk13*	Whole Body/ Lung/Liver/ Kidney/Heart	Null mutant–developmental delay, growth retardation, underdeveloped lungs, liver and kidneys. Embryo lethal due to heart defect	[45]
*Cdk14*			
*Cdk15*			
*Cdk16/PCTAIRE1*	Testes/Ovaries	cKO–failure of spermatogenesis. Female fertility normal	[46]
*Cdk17/PCTAIRE2*	Whole Body/ Brain	Exon deletion–partial embryonic lethality. Increased vertical activity	*
*Cdk18/PCTAIRE3*	Brain	Exon deletion–abnormal behavioural responses to light	*
*Cdk19*			
*Cdk20*			

Blank rows indicate a lack of experimental data for the indicated gene, to the best of our knowledge. * Mouse model is available from the International Mouse Phenotyping Consortium, but a referenced embryonic study could not be found.

**Table 2 ijms-21-05343-t002:** Phenotypes of CDKI (cyclin-dependent kinase inhibitors) KO and cKO in mice.

CDKI	Protein	Developmental Region	KO Phenotypes	References
*Cdkn1a*	p21^CIP1^	Whole Body	KO–no spontaneous tumorigenesis	[47]
		Brain	KO–increased neural migration and differentiation in response to brain injury	[48]
		Intestine	KO–increased susceptibility to intestinal tumours	[49]
		Immune System	KO–no altered susceptibility to retrovirally induced lymphatic tumours	[50]
		Immune System	cKO in T cells–hyposensitive to programmed cell death in response to radiation-induced DNA damage	[51]
*Cdkn1b*	p27^KIP1^	Whole Body/Endocrine System/Ovaries	KO–oversized with larger internal organs, especially the pituitary, and female sterility due to impairment of the formation of corpora lutea	[52]
		Whole Body/ Endocrine System/Ovaries	KO–oversized due to increased cell number and proliferation. Females infertile due to impaired luteal cell differentiation reflecting a disturbance in the hypothalamic–pituitary–ovarian axis	[53]
		Whole Body/ Immune System/ Endocrine System/Eyes/ Ovaries	KO–oversized, especially in thymus, pituitary and adrenal glands and gonadal organs. Disturbed organization of retinas. Female sterility due to impaired development of ovarian follicles	[54]
		Endocrine System/Intestine/ Lung	KO–increased susceptibility to pituitary and intestinal tumours, lesions of the female reproductive tract and lung adenomas following exposure to carcinogens	[55]
		Heart/Ovaries	KO–protective delay of ageing and apoptosis of heart due to ovariectomy by upregulating antioxidant enzymes	[56]
		Testes	KO–decreased testosterone production, despite increased proliferation of testosterone-producing cells	[57]
		Brain	KO–altered neuron migration and differentiation	[58]
		Brain	KO–increased neural progenitor cell proliferation under basal and injury conditions	[59]
		Brain	KO–increased cell proliferation of lower layer neurons	[60]
		Immune System	KO–increased susceptibility to retrovirally induced lymphatic tumours. Increased spleen proliferation	[50]
		Immune System	KO–developed spontaneous T cell lymphomas	[61]
		Eyes	KO–increased cell proliferation and poor repair response after photoreceptor damage	[62]
*Cdkn1c*	p57^KIP2^	Whole Body/ Intestine/ Kidney/Spine/ Endocrine System/Eyes	KO–locational defects in small intestine and muscle. Neonatal lethality due to breathing difficulty from cleft palate. Undersized kidneys, delayed bone development and enlarged adrenal gland. Increased lens cell proliferation and apoptosis	[63]
		Whole Body/ Intestine	KO–neonatal lethality, cleft palate, gastrointestinal abnormalities and short limbs. Increased apoptosis	[64]
		Whole Body/ Spine/Testes/ Ovaries	KO–cleft palate and bone malformations. Most died within 24 hours of birth due to difficulties breathing. Surviving mice showed severe growth retardation. Both sexes were sexually immature	[65]
		Whole Body/ Brain	KO–oversized embryo and macrocephaly, with increased cell proliferation, particularly of neuronal precursors and lower layer neurons. Decreased cell cycle length and increased cell cycle exit at days E14.5 and E16.5	[60]
		Kidney	KO–normal kidney differentiation and development	[66]
		Whole Body/ Spine	cKO maternally–skeletal deformities, reduced body size and perinatal lethality	[67]
		Whole Body	cKO maternally–perinatal lethality, overgrowth during early gestation and weaning, but constrained growth in late gestation	[68]
		Brain	cKO paternally–decreased brain size and reduced number of neural progenitor cells during development	[69]
*Cdkn2a*	p16^INK4a^	Whole Body/ Testes/Ovaries	KO–no morphological or behavioural malformations. No significant increase in spontaneous tumours. Fertile	[70]
		Brain	KO–increased susceptibility to CNS tumours and hydrocephaly following carcinogenic exposure	[71]
		Immune System	KO–normal development, but with thymic hyperplasia	[72]
		Heart	KO–increased cardiomyocyte proliferation and cardiac repair	[73]
		Kidney	KO–increased kidney cell proliferation and decreased apoptosis in response to ischemia-reperfusion injury	[74]
		Lung	KO–reduced inflammation in response to cigarette smoke, but no change to proinflammatory cytokine response	[75]
		Lung	cKO in lung epithelia–reduction in inflammation response to cigarette smoke and suppression of proinflammatory cytokines, but no protective effect	[75]
		Lung	cKO in pleura–no development of malignant mesotheliomas	[76]
		Spine	cKO in intervertebral disc–no change in senescence, reduced apoptosis and altered matrix homeostasis	[77]
*Cdkn2b*	p15^INK4b^	Whole Body/ Testes/Ovaries/ Immune System	KO–viable and fertile with no morphological or behavioural abnormalities at birth. After two months of age, mice had atypical haematopoiesis and increased lymphocytes in the spleen	[78]
*Cdkn2c*	p18^INK4c^	Whole Body/ Endocrine System/ Immune System	KO–gigantism and organomegaly, particularly of the pituitary, spleen and thymus, due to altered cell cycle control. Altered cell cycle entry of resting B cells	[79]
		Whole Body/ Testes/Ovaries/ Immune System/ Endocrine System	KO–viable and fertile with no morphological or behavioural abnormalities at birth. From six months of age mice developed splenomegaly, enlarged lymph nodes and increased incidence of pituitary tumours	[78]
		Endocrine System/Immune System	KO–spontaneous pituitary and lymphatic tumours. Also observed in heterozygous mice, implying haploinsufficiency	[80]
*Cdkn2d*	p19^INK4d^	Whole Body/ Testes	KO–normal growth with no obvious malformations. Decreased, but not total loss of, male fertility due to testicular atrophy and increased apoptosis of germ cells	[81]
		Immune System	KO–increase in mean ploidy level in bone marrow	[82]
		Ears	KO–aberrant cell cycle entry and subsequent apoptosis in sensory hairs leading to hearing loss	[83]
*Cdkn3*		Brain	Exon deletion–decreased grip strength	*

* Mouse model is available from the International Mouse Phenotyping Consortium, but a referenced study could not be found.

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
