# Peer review of "The Role of CDKs and CDKIs in Murine Development"

_ijms, 2020, doi:10.3390/ijms21155343_

Round 1
Reviewer 1 Report
Comments:
The work compiled by the authors is very interesting and will facilitate the CDK research globally. I suggest acceptance with minor revision.
1) The manuscript should be checked throughly for the typographical errors.
2) It become more interesting to the readers if the authors include the structural details of the CDKI as PDB contain lot of information regarding this class of proteins. The details can be represented in the form of the figures or tables.
Author Response
The work compiled by the authors is very interesting and will facilitate the CDK research globally. I suggest acceptance with minor revision.
1) The manuscript should be checked thoroughly for the typographical errors.
Typos within the manuscript have been corrected where found.
2) It become more interesting to the readers if the authors include the structural details of the CDKI as PDB contain lot of information regarding this class of proteins. The details can be represented in the form of the figures or tables.
We thank the reviewer for this suggestion and agree that it is an interesting discussion point. The scope of this review is focussed on the developmental role of these families, rather than protein structure. We have provided references to studies and reviews of the structures of these proteins in the introduction [Lines 29-30], which will direct readers to a series of structure orientated reviews.
Reviewer 2 Report
This paper presents a relatively comprehensive review of CDK biology and definitely adds value to the field.
There are a few things that need to be addressed before this paper can go forward.
In the text, I would have expected a section on CDK inhibitors and what the coverage of CDKs is currently. This does not need to be comprehensive, but should at least include clinical compounds and some other narrow spectrum inhibitors. I would expect to see 2-3 paragraphs as a subsection (after the introduction or just before the discussion) in addition to the table provided.
The tables are not completed, while some of these CDKs are thin on the ground. There is phenotype data for aleast 3 of them that is not included in the paper – have a look on - https://www.mousephenotype.org/
CDK7 - in progress
CDK9 - nothing
CDK10 - in progress
CDK17 - available
CDK18 - available
CDK19 - available
CDK20 - in progress
Cdkn3 – available
The current directions and limitations of the field need to be more clearly and comprehensively described in the discussion.
More minor things -
Figure 1 needs some work, the colour scheme and annotation. It would be nice to have sharper image/overall figure
line 279-291 text format error
line 477, 487, 610 - typo error
There are several other errors/typos through out the text that needs to be cleaned up.
Author Response
This paper presents a relatively comprehensive review of CDK biology and definitely adds value to the field.
We would like to thank the reviewer for their helpful and constructive feedback.
There are a few things that need to be addressed before this paper can go forward.
In the text, I would have expected a section on CDK inhibitors and what the coverage of CDKs is currently. This does not need to be comprehensive, but should at least include clinical compounds and some other narrow spectrum inhibitors. I would expect to see 2-3 paragraphs as a subsection (after the introduction or just before the discussion) in addition to the table provided.
We thank the reviewer for this interesting point and agree that the body of work regarding pharmacological CDK inhibitors is of great importance. As the review focusses on developmental biology, a lot of this work is beyond its scope. To this end, we have included a paragraph [Lines 328-336] that discusses their significance and offers several citations of CDK inhibitor focussed reviews.
The tables are not completed, while some of these CDKs are thin on the ground. There is phenotype data for aleast 3 of them that is not included in the paper – have a look on - https://www.mousephenotype.org/
Thank you for drawing our attention to these works. We have searched for more phenotype data to be included in the tables. Where phenotypic data is available on the mouse website kindly provided, but we could not find a peer-reviewed paper, we have mentioned the data in the table but have made an annotation specifying a lack of published work – “*Mouse model is available from the International Mouse Phenotyping Consortium, but a referenced study could not be found“ [Lines 307-308 and 313-314].
CDK7 - in progress
We are unable to find a paper with a developmental study of Cdk7 function. Embryonic stem cell data is available at mousephenotype.org, which we have not included at this time.
CDK9 – nothing
We are unable to find a paper with a developmental study of Cdk9 function. Phenotypic mouse data exists on mousephenotyping.org which we have included in Table 1.
CDK10 - in progress
We are unable to find a paper with a developmental study of Cdk10 function. Embryonic stem cell data is available at mousephenotype.org, which we have not included at this time.
CDK17 – available
We are unable to find a paper with a developmental study of Cdk17 function. Phenotypic mouse data exists on mousephenotyping.org which we have included in Table 1.
CDK18 – available
We are unable to find a paper with a developmental study of Cdk18 function. Phenotypic mouse data exists on mousephenotyping.org which we have included in Table 1.
CDK19 – available
We are unable to find a paper with a developmental study of Cdk19 function. Phenotypic mouse data exists on mousephenotyping.org but is only available for a heterozygous mouse, therefore it was not included.
CDK20 - in progress
We are unable to find a paper with a developmental study of Cdk20 function. A mouse model exists on mousephenotyping.org but no phenotypic data is available.
Cdkn3 – available
We are unable to find a paper with a developmental study of Cdkn3 function. Phenotypic mouse data exists on mousephenotyping.org which we have included in Table 2.
The current directions and limitations of the field need to be more clearly and comprehensively described in the discussion.
The discussion has been expanded to more explicitly discuss these points [Lines 328-336, 432-433 and 346-350].
More minor things -
Figure 1 needs some work, the colour scheme and annotation. It would be nice to have sharper image/overall figure
Figure 1 has been amended for clarity. The colour scheme was altered to increase sharpness and contrast, and organ labels were added.
line 279-291 text format error
This paragraph has been reformatted to conform with the rest of the text.
line 477, 487, 610 - typo error
These typo errors in the references’ DOIs have been resolved.
There are several other errors/typos through out the text that needs to be cleaned up.
Typos within the manuscript have been corrected where found.
Round 2
Reviewer 2 Report
The authors have addressed my concerns, after a final proof read this review is ready to go.